# Scaling neural machine translation to 200 languages

NLLB Team*

The development of neural techniques has opened up new avenues for research in machine translation. Today, neural machine translation (NMT) systems can leverage highly multilingual capacities and even perform zero-shot translation, delivering promising results in terms of language coverage and quality. However, scaling quality NMT requires large volumes of parallel bilingual data, which are not equally available for the 7,000+ languages in the world[1]. Focusing on improving the translation qualities of a relatively small group of high-resource languages comes at the expense of directing research attention to low-resource languages, exacerbating digital inequities in the long run. To break this pattern, here we introduce No Language Left Behind—a single massively multilingual model that leverages transfer learning across languages. We developed a conditional computational model based on the Sparsely Gated Mixture of Experts architecture[2–7], which we trained on data obtained with new mining techniques tailored for low-resource languages. Furthermore, we devised multiple architectural and training improvements to counteract overfitting while training on thousands of tasks. We evaluated the performance of our model over 40,000 translation directions using tools created specifically for this purpose—an automatic benchmark (FLORES-200), a human evaluation metric (XSTS) and a toxicity detector that covers every language in our model. Compared with the previous state-of-the-art models, our model achieves an average of 44% improvement in translation quality as measured by BLEU. By demonstrating how to scale NMT to 200 languages and making all contributions in this effort freely available for non-commercial use, our work lays important groundwork for the development of a universal translation system.

The recent advent of neural machine translation (NMT) has pushed translation technologies to new frontiers, but its benefits are unevenly distributed[1]. The vast majority of improvements made have mainly benefited high-resource languages, leaving many low-resource languages behind. (For the purpose of our research, we define a high-resource language as a language for which we have at least 1 million sentences of aligned textual data (or bitext) with another language). This disparity could largely be attributed to a data gap: NMT models typically require large volumes of data to produce quality translations and, by definition, these volumes are not available for lower-resource languages. The No Language Left Behind (NLLB-200) project seeks to overcome this limitation by leveraging previously unknown approaches for building massively multilingual models with cross-lingual transfer abilities[8,9], thereby enabling related languages to learn from each other[1,10,11].

It has now been widely acknowledged that multilingual models have demonstrated promising performance improvement over bilingual models[12]. However, the question remains whether massively multilingual models can enable the representation of hundreds of languages without compromising quality. Our results demonstrate that doubling the number of supported languages in machine translation and maintaining output quality are not mutually exclusive endeavours. Our final model—which includes 200 languages and three times as

many low-resource languages as high-resource ones—performs, as a mean, 44% better than the previous state-of-the-art systems. This paper presents some of the most important data-gathering, modelling and evaluation techniques used to achieve this goal.

First, compared with their high-resource counterparts, training data for low-resource languages are expensive and logistically challenging to procure[13–15]. Publicly available digital resources are either limited in volume or difficult for automated systems to detect (particularly in large public web datasets such as CommonCrawl). Regardless of whether collecting a critical mass of human-translated seed data is necessary, sufficient data acquisition relies on large-scale data mining and monolingual data pipelines[16–19]. The latter techniques are often affected by noise and biases, thereby making validating the quality of the datasets they generate tedious[20]. In NLLB-200, we show that a distillation-based sentence encoding technique, LASER3 (ref. 21), facilitates the effective mining of parallel data for low-resource languages.

Second, on the modelling side, we use an assemblage of seed, mined, open-source and back-translated datasets to train multilingual conditional computational models (more specifically, Sparsely Gated Mixtures-of-Experts models[2–7] that enable cross-lingual transfer between related languages without increasing interference between unrelated languages). We show how we can achieve state-of-the-art

performance with a more optimal trade-off between cross-lingual transfer and interference, and improve performance for low-resource languages.

Finally, for the purpose of quality evaluation, we created FLORES-200—a massive multilingual benchmark that enables the measurement of translation quality across any of the approximately 40,000 translation directions covered by the NLLB-200 models. Apart from automatic metrics, we also created Cross-lingual Semantic Text Similarity (XSTS) and Evaluation of Toxicity (ETOX). XSTS is a human evaluation protocol that provides consistency across languages; ETOX is a tool to detect added toxicity in translations using toxicity word lists.

Beyond creating these models, we also reflect on the potential societal impact of NLLB. To amplify the practical applicability of our work in service of low-resource-speaking communities, we provide all the benchmarks, data, code and models described in this effort as resources freely available for non-commercial use (https://github.com/facebookresearch/fairseq/tree/nllb) (see Data and Code availability statements for details).

## Automatically creating translation training data

The current techniques used for training translation models are difficult to extend to low-resource settings, in which aligned bilingual textual data (or bitext data) are relatively scarce[22]. Many low-resource languages are supported only by small targeted bitext data consisting primarily of translations of the Christian Bible[23], which provide limited domain diversity.

To build a large-scale parallel training dataset that covers hundreds of languages, our approach centres around extending existing datasets by first collecting non-aligned monolingual data. Then, we used a semantic sentence similarity metric to guide a large-scale data mining effort aiming to identify sentences that have a high probability of being semantically equivalent in different languages[18].

### Language identification for monolingual data collection

Collecting monolingual data at scale requires a language identification (LID) system that accurately classifies textual resources for all NLLB-200 languages. Although LID could be seen as a solved problem in some domains[24], it remains an open challenge for web data[25,26]. Specifically, issues coalesce around domain mismatch[26], similar language disambiguation[27] and successful massively multilingual scaling[28].

Devoted attention to advancing LID techniques led to a noticeable increase in both language coverage and accuracy over time. CLD3 (https://github.com/google/cld3) and fasttext[29] are two readily available models offering high detection performance for 107 and 187 languages, respectively. By using numerous public datasets, previous studies[30,31] report even higher coverage—464 and 1,366 languages, respectively. Another study[32] scales LID performance up to 1,629 languages using word lists and self-supervision to bootstrap training data found on the web. However, these approaches using found data suffer from domain imbalance. That is, because the available text domains vary by language, classifiers conflate different domains with different languages.

In our work, we curated FLORES-200 to use as a development set so that our LID system performance[33] is tuned over a uniform domain mix. Our approach combines a data-driven fasttext model trained on FLORES-200 with a small set of handwritten rules to address human feedback on classification errors. These rules are specifically mentioned in section 5.1.3 of ref. 34 and include linguistic filters to mitigate the learning of spurious correlations due to noisy training samples while modelling hundreds of languages.

We compare our LID model with three publicly available models: CLD3, LangId (https://github.com/saffsd/langid.py) and LangDetect (https://pypi.org/project/langdetect/). Table 1 reports the performance

**Table 1 | Comparison of publicly available language identification models with various intersections of labels**

| No. of supported languages | | FLORES-200 ∩ CLD3 ∩ LangId ∩ LangDetect | | FLORES-200 ∩ CLD3 ∩ LangId | | FLORES-200 ∩ CLD3 | |
|---|---|---|---|---|---|---|---|
| | | 51 labels | | 78 labels | | 95 labels | |
| | | F1 | FPR | F1 | FPR | F1 | FPR |
| LangDetect | 55 | 97.3 | 0.0526 | 64.4 | 0.4503 | 53.1 | 0.4881 |
| LangId | 97 | 98.6 | 0.0200 | 92.0 | 0.0874 | 75.8 | 0.2196 |
| CLD3 | 107 | 98.2 | 0.0225 | 97.7 | 0.0238 | 97.0 | 0.0283 |
| Ours | **218** | **99.4** | **0.0084** | **98.8** | **0.0133** | **98.5** | **0.0134** |

F1 is the micro-F1 score, and FPR is the micro-false-positive rate.

on three cascading sets of languages intersecting with NLLB-200: (1) 51 languages also supported by LangId, LangDetect and CLD3; (2) 78 languages also supported by LangId and CLD3; (3) 95 languages also supported by CLD3. We also report false-positive rates (FPR) to reflect the impact of false positives on unseen languages. Our results show that our model is equipped to handle all 200 languages found in FLORES-200 while achieving notably higher performance than LangId, LangDetect and CLD3. Furthermore, the gain in F1 score is accompanied by a notable improvement in FPR, suggesting a much stronger fit for extracting low-resource languages from web corpora[32].

### Mining for bitext

Previous work[35] notes that translation quality generally increases with the amount of high-quality training data, which is difficult to procure when working with low-resource languages. Existing parallel corpora for low-resource languages are often conveniently drawn from known multilingual collections, such as the Christian Bible or the publications of multinational organizations, which are limited in quantity and domain. To overcome this problem, we created training datasets through global bitext mining in publicly available web content (drawn from repositories such as CommonCrawl). The underlying idea of our bitext mining approach is first to learn a multilingual sentence embedding space and use a similarity measure in that space to decide whether two sentences are parallel. This comparison can be done for all possible pairs in two collections of monolingual texts.

As our mining approach requires a multilingual embedding space, there are several challenges when scaling this representation to all NLLB-200 languages. First, we had to ensure that all languages were well learnt and that we accounted for large imbalances in available training data. Second, training a massively multilingual sentence encoder from scratch each time a new set of languages is introduced is computationally expensive. Furthermore, the main drawback of this approach is that the learnt embedding spaces from each new model are not necessarily mutually compatible. This can make mining intractable as for each new encoder, the entirety of available monolingual data needs to be re-embedded (for example, for English alone, this means thousands of millions of sentences and considerable computational resources). We solved this problem using a teacher–student approach[21] that extends the LASER embedding space[36] to all NLLB-200 languages. Languages are trained either as individual students or together with languages from the same family. The training of students follows the approach described in ref. 21.

Our approach enables us to focus on the specifics of each language while taking advantage of related languages, which is crucial for dealing with very low-resource languages. (A language is defined as very low-resource if it has fewer than 100,000 samples across all pairings with any other language in our dataset). Using this method, we generated more than 1,100 million new sentence pairs of training data for 148 languages. This additional training data, paired with back translation

**Table 2 | Improvements from EOM and CL**

| | eng_Latn-xx | | | | xx-eng_Latn | | | | xx-yy | Average |
|---|---|---|---|---|---|---|---|---|---|---|
| | **All** | **High** | **Low** | **Very low** | **All** | **High** | **Low** | **Very low** | **All** | **All** |
| **(1)** Baseline MoE | 44.8 | 54.3 | 41.4 | 39.0 | 56.2 | 64.0 | 53.4 | 52.5 | 41.9 | 47.6 |
| **(2)** Baseline MoE+ CL | 45.2 | **54.7** | **41.8** | 39.5 | **57.6** | **64.5** | 55.1 | 55.4 | 42.7 | 48.5 |
| **(2)** Baseline MoE+CL+EOM | **45.4** | 52.9 | 41.6 | **41.2** | 57.2 | 61.4 | 55.1 | **56.4** | **44.9** | **51.0** |

We report chrF++ scores on FLORES-200 dev set on different types of language pairs. For eng_Latn-xx and xx-eng_Latn, we included all 199 pairs. For xx-yy, we randomly chose 200 directions. We observe that combining EOM and CL is particularly helpful for low and very low-resource languages. A language is defined as a very low resource if it has fewer than 100,000 samples across all pairings with any other language in our dataset. The highest score in each column is shown in bold.

(a conventional technique for data augmentation in NMT; ref. 37), ushered notable improvements in translation quality—specifically, +12.5 chrF++ (ref. 38) for translating very low-resource languages into English. For more details, see Supplementary Information D.

## Modelling

Even with marked data volume increases, the main challenge of low-resource translation is for training models to adequately represent 200 languages while adjusting to variable data capacity per language pair. Apart from techniques such as data augmentation (for example, with back translation) and self-supervision strategies on monolingual data, we used conditional computational models—more specifically, Sparsely Gated Mixture of Experts (henceforth MoE)—to minimize interference between unrelated language directions.

MoE transformer models differ from dense transformer models in that some of the feed-forward network layers are replaced with MoE layers in both the encoder and the decoder. An MoE layer consists of $E$ experts (each is a feed-forward network) and a gating network to decide how to route input tokens to experts. The transformer encoder–decoder model, supplemented with MoE layers and their respective gating networks, learns to route input tokens to the corresponding top two experts by optimizing a linearly weighted combination of label-smoothed cross entropy[39] and an auxiliary load balancing loss[6].

We find that vanilla MoE models with overall dropout are suboptimal for low-resource languages and significantly overfit on low-resource pairs. To remedy this issue, we designed Expert Output Masking (EOM), a regularization strategy specific to MoE architectures, and compared it with existing regularization strategies, such as Gating Dropout[40]. We find that Gating Dropout performs better than vanilla MoE with overall dropout but is outperformed by EOM.

To further reduce overfitting on low-resource language pairs, we devised a curriculum learning that introduces language pairs in phases during model training. Pairs that empirically overfit within $K$ updates are introduced with $K$ updates before the end of training. This reduces overfitting while allowing pairs that benefit from additional training

to continue their learning. Table 2 shows that combining curriculum learning and EOM improves performance, especially on low and very low-resource language pairs (see section 'Modelling' for more details).

To understand how MoE models are helpful for multilingual machine translation, we visualize similarities of experts in the MoE layers using heat maps (Fig. 1a–d). These heat maps demonstrate that in late decoder layers (Fig. 1d), languages are being separated (that is, dispatched to different sets of experts). Moreover, we observe that languages within the same family are highly similar in their choice of experts (that is, the late decoder MoE layers are language-specific). This is particularly the case for the Arabic dialects (the six rows and columns in the top-left corner), languages in the Benue–Congo subgrouping, as well as languages in the Devanagari script. By contrast, the early decoder MoE layers (Fig. 1c) seem to be less language-specific. The late encoder MoE layers are particularly language-agnostic in how they route tokens as can be attested by the uniform heat map in Fig. 1b.

Combining data (see section 'Automatically creating translation training data') and modelling contributions, Table 3 shows that NLLB-200 outperforms the nearest state-of-the-art system by almost +7.3 spBLEU (ref. 41) on average, constituting a 44% improvement. We then compared NLLB-200 with a few other state-of-the-art models, such as Deepnet[42] and M2M-100 (ref. 1), to report scores for 87 languages against FLORES-101. On this smaller subset, NLLB-200 again outperforms by +7.0 spBLEU on average. Overall, the results show that NLLB-200 improves on state-of-the-art systems by a notable margin despite supporting 200 languages, or twice as many languages (and more than 30,000 additional directions) compared with any previous work. We also show in additional experiments that NLLB-200 is a general-purpose NMT model, transferable to other domains by fine-tuning on small quantities of high-quality bitexts (see Supplementary Information E.3).

## Evaluations

Among the many aspects of model performance that can be evaluated[43], this section emphasizes three aspects that have a marked impact on

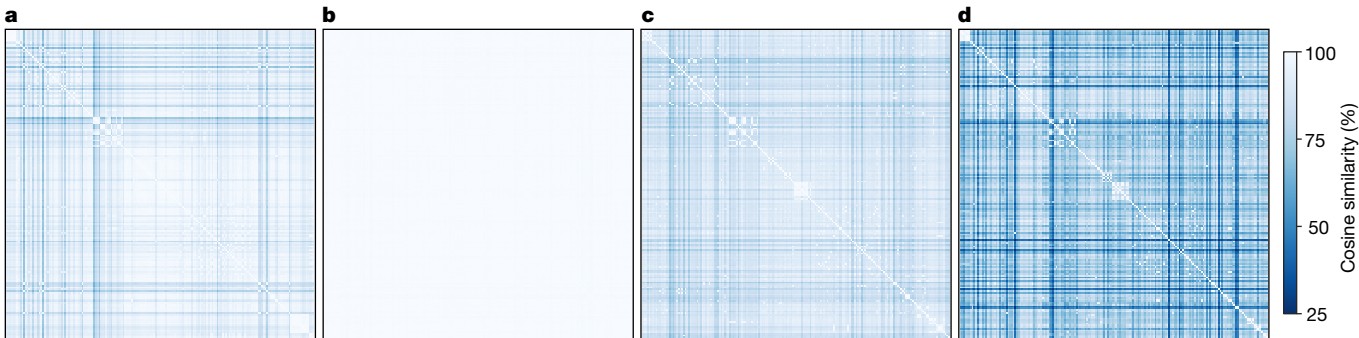

**Fig. 1 | Cosine similarity scores between languages at different layers of the encoder–decoder architecture. a–d**, The first (**a**) and last (**b**) encoder layers and then the first (**c**) and last (**d**) decoder layers. The similarity is measured with respect to the gating decisions (expert choice) per language (source side in the encoder and target side in the decoder). Lighter colours represent higher experts similarity, hence, a language-agnostic processing.

**Table 3 | Comparison of FLORES-101 devtest**

| | eng_Latn-xx | xx-eng_Latn | xx-yy | Average |
|---|---|---|---|---|
| **87 languages** | | | | |
| M2M-100 | –/– | –/– | –/– | 13.6/– |
| Deepnet | –/– | –/– | –/– | 18.6/– |
| NLLB-200 | **35.4**/52.1 | **42.4**/62.1 | **25.2**/43.2 | **25.5**/43.5 |
| **101 languages** | | | | |
| DeltaLM | 26.6/– | 33.2/– | 16.4/– | 16.7/– |
| NLLB-200 | **34.0**/50.6 | **41.2**/60.9 | **23.7**/41.4 | **24.0**/41.7 |

We evaluated using FLORES-101 for 10,000 directions. We report both spBLEU and chrF++ scores when available. Scores for DeltaLM are taken from the FLORES-101 leaderboard. M2M-100 and Deepnet averages only apply to 87 languages that overlap with FLORES-101. The performance of NLLB-200 was evaluated on this subset of languages. The highest score in each column and in each grouping of languages is shown in bold.

the overall quality assessment: benchmarks for automatic evaluation, human evaluation protocols and toxicity evaluation.

## A benchmark for automatic evaluation using FLORES-200

The quality of NMT outputs is typically evaluated by automatic metrics such as BLEU[44] or spBLEU[41]. The computation of automatic quality scores using these metrics requires benchmark datasets that provide gold-standard human translations as references. In turn, the apples-to-apples evaluation of different approaches made possible by these benchmark datasets gives us a better understanding of what requires further research and development. For example, creating benchmark data sets at the Workshop on Machine Translation (WMT)[45] led to rapid progress in translation directions such as English to German and English to French.

For massively multilingual NMT, the largest benchmark dataset available was FLORES-101, which supports roughly half the number of languages in NLLB-200. The necessary expansion of FLORES-101 to FLORES-200 constitutes a further challenge in terms of quality assurance, in part because of differences in standardization practices and limited access to professional translators for all languages involved. To overcome this challenge, we adapted our workflow to pay particular attention to quality assurance mechanisms. The FLORES-200 workflow consists of four phases: (1) alignment; (2) translation, initial quality assurance and iteration(s); (3) final quality assurance; and (4) completion. A language FLORES-200 set is considered ready after passing a final human quality test with a 90 out of 100 quality score (that is, independent raters agreed with 90% of the FLORES-200 reference translations in that direction).

As a result of this redesigned workflow, we produced a three-split (dev, devtest, test) data set of parallel human reference translations for all NLLB-200 languages meeting the 90% quality threshold in a maximum turnaround time of 287 days (119 days on average, 70 days minimum). (Note that to avoid leakage with our models, we filtered data from FLORES and other evaluation benchmarks used (such as WMT and IWSLT) from our training data. This was done by comparing

the hashes of training sentences against those of evaluation sentences, using the xxHash algorithm). Please refer to Supplementary Information C for more details on the evaluation process. Figure 2 shows the quality scores for all languages, some of which are labelled as examples.

## Reliable human evaluation

State-of-the-art automatic metrics often fail to capture aspects of language that, while subtle, can have a notable bearing on translation quality. Human evaluations are, therefore, essential to ensuring meaningful quality assessments[46]. That said, relying on them comes with two challenges: (1) any large-scale human evaluation of NMT quality, regardless of the number of translation directions involved, contends with potentially low inter-evaluator agreement (in the vicinity of 0.5 kappa); and (2) massively multilingual NMT introduces another complexity—that of quality evaluation consistency across language directions. We address these two issues by developing XSTS[47], a new scoring metric focused on meaning, and by using a protocol that allows for the calibration of scores across evaluators and language pairs.

XSTS is a human evaluation protocol inspired by STS[48], emphasizing meaning preservation over fluency. XSTS uses a five-point scale, in which 1 is the lowest score, and 3 represents the acceptability threshold. To ensure consistency not only across languages but also among different evaluators of any given language, we included the same subset of sentence pairs in the full set of sentence pairs given to each evaluator, making it possible to calibrate results.

We find that automated metrics such as spBLEU and chrF++ correlate reasonably well with calibrated human evaluations of translation quality, as shown in Fig. 3. Spearman's $R$ correlation coefficients between aggregated XSTS and spBLEU, chrF++ (corpus) and chrF++ (average sentence-level) are 0.710, 0.687 and 0.694, respectively. Other correlation coefficients (Kendall's $\tau$ and Pearson's $R$) have the same ordering. Corpus spBLEU provides the best nominal correlation, followed by average sentence-level chrF++.

We also find that calibrated human evaluation scores correlate more strongly with automated scores than uncalibrated human evaluation scores across all automated metrics and choices of correlation coefficient. In particular, uncalibrated human evaluation scores have a Spearman's $R$ correlation coefficient of 0.625, 0.607 and 0.611 for spBLEU, chrF++ (corpus) and chrF++ (average sentence-level), respectively.

Overall, a sample of 55 language directions were evaluated, including 8 into English, 27 out of English, and 20 other direct language directions. The overall mean of calibrated XSTS scores was 4.26, with 38/55 directions scoring over 4.0 (that is, high quality) and 52/56 directions scoring over 3.0.

We hypothesize that added toxicity may be because of the presence of toxicity in the training data and used our detectors to estimate, more specifically, unbalanced toxicity in the bitext data. We find that estimated levels of unbalanced toxicity vary from one corpus of bitext to the next and that unbalanced toxicity can be greatly attributed to misaligned bitext. In other words, training with this misaligned bitext could encourage mistranslations with added toxicity.

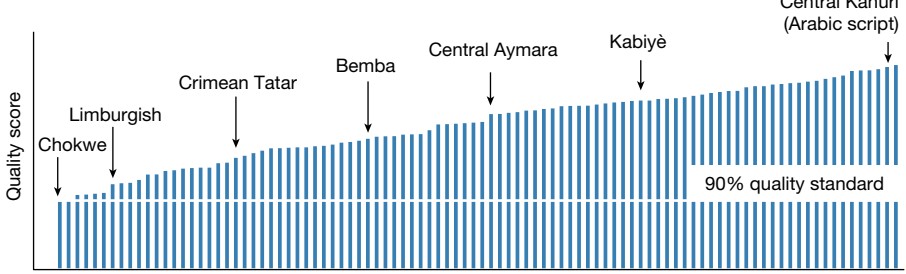

**Fig. 2 | Quality of FLORES-200.** Quality assurance scores for the languages in FLORES-200. The minimum acceptable standard is 90%.

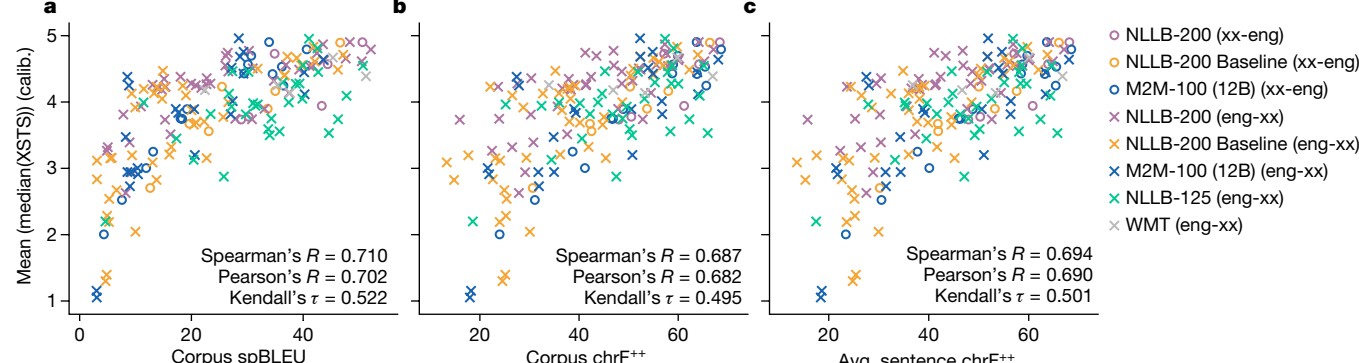

**Fig. 3 | Correlations between aggregated human quality scores and automated metrics. a**, The relationship between spBLEU and XSTS. **b**, The relationship between chrF++ and XSTS. **c**, The relationship between average sentence-level chrF++ and XSTS. All automated scores were computed only on the sentences evaluated for a given model and translation direction (either the full FLORES-200 dataset or a subset). NLLB-200 refers to a 55B parameter MoE model, and NLLB-200 Baseline refers to a dense 3.3B parameter model.

To mitigate this issue, we designed a bitext filtering procedure based on the detection of multiple instances of added toxicity (that is, cases in which one sentence in the bitext pair contains at least two more toxic items than the other sentence in the pair). (A previous detector quality analysis showed that a higher precision was reached in this situation). We added this toxicity filtering procedure as an option to the filtering process and experimented with or without it for comparison.

The experimental results on the FLORES-200 dev set for 10 translation directions (from and into English for Somali, Southern Sotho, Twi, Umbundu and Venetian) show that after filtering an average amount of around 30% parallel sentences, the translation quality (chrF++) improves by 5% and added toxicity (ETOX) reduces by the same amount. Therefore, the filtering pipeline that includes toxicity filtering not only reduces the number of toxic items in the translation output but also improves the overall translation performance.

## Conclusion

In 2016, the United Nations declared internet access a basic human right. Although the intent of this declaration was to limit censorship and allow for information and ideas to flow without interference, much of the internet today remains inaccessible to many due to language barriers. Our effort was designed to contribute one solution to help alter this status quo.

For many low-resource language communities, NLLB-200 is one of the first models designed to support translation into or out of their languages. Although applications of these new translation capabilities could be found in several domains of everyday life, we believe their impact would be most significant in a domain such as education. In formal educational settings, for instance, students and educators belonging to low-resource language groups could, with the help of NLLB-200, tap into more books, research articles and archives than before. Within the realms of informal learning, low-resource language speakers could experience greater access to information from global news outlets and social media platforms, as well as online encyclopaedias such as Wikipedia. Access to machine translation motivates more low-resource language writers or content creators to share localized knowledge or various aspects of their culture. Giving individuals access to new translation tools could thus open up opportunities for bidirectional learning, thereby also challenging Western-centric modes of knowledge production and dissemination, ultimately aiding in revitalizing certain minority cultures and languages.

Since launching NLLB-200, we can already see the impact of the model across many directions. Four months after the launch of NLLB-200, Wikimedia reported that our model was the third most used machine translation engine used by Wikipedia editors (accounting for 3.8% of all published translations) (https://web.archive.org/web/20221107181300/https://nbviewer.org/github/wikimedia-research/machine-translation-service-analysis-2022/blob/main/mt_service_comparison_Sept2022_update.ipynb). Compared with other machine translation services and across all languages, articles translated with NLLB-200 has the lowest percentage of deletion (0.13%) and highest percentage of translation modification kept under 10%.

In many ways, the composition of the NLLB-200 effort speaks to the centrality of interdisciplinarity in shaping our vision. Machine translation and AI advancements lie at the intersection of technological, cultural and societal development, and thus require scholars with diverse training and standpoints to fully comprehend every angle[49,50]. It is our hope that in future iterations, NLLB-200 continues to include scholars from fields underrepresented in the world of machine translation and AI, particularly those from humanities and social sciences backgrounds. More importantly, we hope that teams developing these initiatives would come from a wide range of race, gender and cultural identities, much like the communities whose lives we seek to improve.

Finally, we want to emphasize that overcoming the challenges that prevent the web from being accessible to speakers of all languages requires a multifaceted approach. At the technical level, NLLB-200 overcomes many data, modelling and evaluation challenges in NMT research, but it still has its limitations, some of which are documented in Supplementary Information G. As a single technological intervention, NLLB-200 is all but one piece of a massive puzzle; policy interventions aimed at more fundamental issues surrounding education, internet access and digital literacy are imperative to eradicate the structural problem of language disparities.

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

**NLLB Team**

**Marta R. Costa-jussà**[1✉], **James Cross**[2], **Onur Çelebi**[1], **Maha Elbayad**[3], **Kenneth Heafield**[4], **Kevin Heffernan**[4], **Elahe Kalbassi**[3], **Janice Lam**[3], **Daniel Licht**[3], **Jean Maillard**[3], **Anna Sun**[3], **Skyler Wang**[3,5], **Guillaume Wenzek**[1], **Al Youngblood**[3], **Bapi Akula**[3], **Loic Barrault**[1], **Gabriel Mejia Gonzalez**[3], **Prangthip Hansanti**[3], **John Hoffman**[3], **Semarley Jarrett**[3], **Kaushik Ram Sadagopan**[3], **Dirk Rowe**[3], **Shannon Spruit**[1], **Chau Tran**[3], **Pierre Andrews**[1], **Necip Fazil Ayan**[3], **Shruti Bhosale**[3], **Sergey Edunov**[3], **Angela Fan**[2], **Cynthia Gao**[3], **Vedanuj Goswami**[3], **Francisco Guzmán**[3], **Philipp Koehn**[2,6], **Alexandre Mourachko**[1], **Christophe Ropers**[3], **Safiyyah Saleem**[2], **Holger Schwenk**[1] & **Jeff Wang**[3]

[1]Foundational AI Research (FAIR), Meta, Paris, France. [2]Foundational AI Research (FAIR), Meta, New York, NY, USA. [3]Foundational AI Research (FAIR), Meta, Menlo Park, CA, USA. [4]Foundational AI Research (FAIR), Meta, London, UK. [5]University of California, Berkeley, CA, USA. [6]Johns Hopkins University, Baltimore, MD, USA.

# Methods

## Data

This section describes the steps taken to design our language identification system and bitext mining protocol.

**Language identification.** To train language identification models, we used fasttext[33,51], which has been widely used for text classification tasks because of its simplicity and speed. We embedded character-level $n$-grams from the input text and leveraged a multiclass linear classifier on top. The lightweight nature of fasttext enables our LID models to handle web-scale data. Furthermore, a linear model has the benefit of being easily explainable, allowing us to trace any classification error back to its root cause. This is instrumental in addressing common pitfalls that arise when detecting language on web corpora[32].

**Classifier design.** We experimented with two different designs. First, we used a combination of multiple binary classifiers in which the final decision was obtained by selecting the language with the highest score after applying a threshold. We applied threshold optimization so that when the confidence of a classifier is low, the corresponding language is not considered for the final decision. A sentence was filtered out if none of the classifiers surpassed its threshold. Second, we built a multiclass classifier using softmax over all possible languages. In this case, the threshold optimization is done after the softmax.

Our results directed us to focus on the second approach, which offers several advantages. First, changing the threshold for one language did not affect the performance of the other (which is not true in the first setting). Second, this approach generalizes better to out-of-domain data, which is our primary use case (Wikipedia → web data). Finally, a single classifier has the added benefit of being computationally simpler, thus streamlining the language identification process.

**Training data and handling massive class imbalance.** We used publicly available datasets to train our LID system, partially covering our languages of interest. The public datasets deployed were mostly built from web pages such as CommonCrawl. We then supplemented these with NLLB-Seed data (Supplementary Information B) for any missing languages. However, this supplementation is insufficient in ensuring balance in the raw training data[32,30]. For example, English alone represents 10.1% of our training data, whereas Minangkabau (Latin script) represents only 0.06%. Following ref. 10, we experimented with multiple settings of temperature upsampling for underrepresented languages, in which sentences from a language $l$ representing $p_l$ per cent of the data set are sampled proportionally to $p_l^{1/T}$. Optimal performance was obtained at $1/T = 0.3$ (for more details, see section 5.1 of ref. 34).

**Training parameters.** Our best-performing model was trained with softmax loss over two epochs with a learning rate of 0.8 and embeddings with 256 dimensions. We discarded words with less than a thousand occurrences after upsampling and selecting a minimum and maximum character $n$-gram length of two and five, respectively (which were assigned a slot in buckets of size 1,000,000). (In fasttext, we refer to 'word' when it is separated by spaces. When it is a non-segmenting language, there is only one 'word' for the whole sentence (and we take character $n$-grams)). All hyperparameters were tuned on FLORES-200 dev (see section 5.1.2 of ref. 34).

**Improving LID with linguistic analysis.** Language identification is a challenging task in which numerous failure modes exist, often exacerbated by the gaps between the clean data on which LID models are trained and noisy data on which LID models are applied. In other words, LID models trained in a supervised manner on fluently written sentences may have difficulty identifying grammatically incorrect and incomplete strings extracted from the web. Furthermore, models can easily learn spurious correlations that are not meaningful for the task itself. Given these challenges, we collaborated closely with a team of linguists throughout different stages of LID development to identify proper focus areas, mitigate issues and explore solutions (see section 5.1.3 of ref. 34).

**Bitext mining.** The overall approach for bitext mining focused on starting with a massively multilingual sentence encoder teacher model and adapting it to several different low-resource student models. This approach enabled us to add low-resource languages without competing with high-resource languages for capacity. Doing so circumvents the need to retrain the entire model from scratch while maintaining compatibility with the multilingual embedding spaces for subsequent mining. Extended data Fig. 1 summarizes the overall architecture of the teacher–student approach. The teacher, LASER2, is an improved version of the open-source LASER encoder (https://github.com/facebookresearch/LASER). The original training procedure[36] was adapted to include SentencePiece tokenization (including a vocabulary of 7,000 tokens) and the upsampling of low-resource languages.

The architecture of the five-layer BiLSTM encoder and the max pooling method to obtain sentence embeddings were left unchanged. The training was then performed on the same 93 languages with public resources obtained from OPUS[52]. See ref. 36 for details on the original LASER training procedure. Training of the students followed the approach described in greater detail in ref. 21, summarized below:
- students specialized in one language or several similar languages;
- students were randomly initialized because we wanted to handle low-resource language for which we did not have a pre-trained language model;
- students may have a dedicated SentencePiece vocabulary different from the teacher to better accommodate scripts and tokens in the student languages;
- as we used cosine distance for bitext mining (Fig. 1), students learnt to minimize the cosine loss with the teacher;
- students can have an MLM loss to leverage student language monolingual data (Fig. 1).

**Training parameters.** Our student encoders used a 12-layer transformer with a hidden size of 1,024, four attention heads, and around 250 million parameters. All students were trained with available bitexts in their respective language, complemented by 2 million sentences of English/English and English/Spanish. The motivation behind this approach is to anchor the students to the English embedding space, increasing robustness by including English/Spanish bitexts from CCMatrix and allowing for the joint learning of new languages. This technique is particularly useful when only limited amounts of bitexts are available to train the students. Teacher–student training was performed on 16 GPUs, the ADAM optimizer, a learning rate of 0.0005 and a batch size of 10,000. We trained student encoders for 148 languages and named these models LASER3.

**Proxy metric for new encoders.** Mined bitexts were subsequently used to improve translation quality for the languages of NLLB-200. However, mining and NMT training are computationally expensive, and it is intractable to perform this evaluation systematically for many different sentence encoder variants. As an evaluation proxy, we used a mining-based multilingual similarity search error rate, referred to here as xsim. In contrast to cosine accuracy, which aligns embeddings based on the highest cosine score, xsim aligns source and target embeddings based on the highest margin score, which has been shown to be beneficial in mining[53]. The margin-based score is defined as

$$
\text{score}(x, y) = \text{margin}\left(\cos(x, y), \sum_{z \in NN_k(x)} \frac{\cos(x, z)}{2k} + \sum_{v \in NN_k(y)} \frac{\cos(y, v)}{2k}\right) \tag{1}
$$

where $x$ and $y$ are the source and target sentences, and $NN_k(x)$ denotes the $k$ nearest neighbours of $x$ in the other language. We set $k$ to 4. All xsim results are calculated on FLORES-200 devtest, using the ratio margin, where $margin(a, b) = a/b$. Moreover, all scores are calculated for translations into English (that is, xxx → eng). English is encoded by the teacher, and the other language is encoded by the LASER3 student. To facilitate further research using xsim, we also provide this evaluation method as an open-source resource (https://github.com/facebookresearch/LASER/).

**End-to-end encoder evaluation.** Once we had identified the best sentence encoder for each language using the xsim scores, we performed mining, added the mined data to the existing bitexts and trained a bilingual NMT system. Initial experiments indicated that a threshold on the margin of 1.06 seems to be the best compromise between precision and recall for most languages. For these NMT baselines, we do not apply extra filtering on the bitexts and leave this to the training procedure of our massively multilingual NMT system.

We did not attempt to optimize the architecture and parameters of the bilingual NMT systems to the characteristics of each language pair but used the same architecture for all. Therefore, the reported results should not be interpreted as the best possible ones given the available resources—they are mainly provided to validate the mined bitexts. We used a 12-layer encoder and decoder and trained for 100 epochs. Moreover, we looked for the best performance on the FLORES-200 development set and report detokenized BLEU on the FLORES-200 devtest.

## Modelling

In this section, we first describe the multilingual machine translation task setup, which includes tokenization and base model architecture. Then, we outline how we leveraged conditional computation for massively multilingual machine translation with EOM regulation and our Curriculum Learning (CL) strategy for low-resource languages.

**Task setup.** We modelled multilingual NMT as a sequence-to-sequence task, in which we conditioned on an input sequence in the source language with an encoder and generated the output sequence in the expected target language with a decoder[54]. With the source sentence $S$, source language $\ell_s$, and target language $\ell_t$ in hand, we trained to maximize the probability of the translation in the target language $T$—that is, $P(T|S, \ell_s, \ell_t)$. Below, we discuss details of the (1) tokenization of the text sequences in the source and target languages; and (2) model architecture with the input and output designed specifically for multilingual machine translation. For further details on the task setup, such as the amount of training data per language pair, please refer to Supplementary Information F or section 8 of ref. 34.

**Segmentation with SentencePiece.** To tokenize our text sequences, we trained a single SentencePiece model (SPM)[55] for all languages. We sampled a total of 100 million sentences from primary bitext data. To ensure low-resource languages are well-represented in the vocabulary, we downsampled high-resource and upsampled low-resource languages with a sampling temperature of five (ref. 10). Notably, vocabulary size is an important hyperparameter in multilingual translation models involving low-resource languages[56–58]. The vocabulary size of our trained SPM model is 256,000. Such a large vocabulary ensures adequate representation across the wide spectrum of languages we support.

**Model architecture.** Our sequence-to-sequence multilingual machine translation model is based on the transformer encoder–decoder architecture[59]. The encoder transforms the source token sequence into a sequence of token embeddings. Then, the decoder attends to the encoder output and autoregressively generates the target sentence token by token. More precisely, the encoder takes the sequence of tokens $W = (w_1, ..., w_S)$ and the source language $\ell_s$, and produces a sequence of embeddings $H = (h_1, ..., h_S)$, which are then provided to the decoder with the target language $\ell_t$ to produce the target tokens $V = (v_1, ..., v_T)$ sequentially. In sum,

$$H = \text{encoder}(W, \ell_s), \tag{2}$$

$$\forall i \in [1, ..., T], \; v_{i+1} = \text{decoder}(H, \ell_t, v_1, ..., v_i). \tag{3}$$

Note that we prefixed the source sequence with the source language, as opposed to the target language, as done in previous work[10,60]. We did so because we prioritized optimizing the zero-shot performance of our model on any pair of 200 languages at a minor cost to supervised performance. Empirically, we find zero-shot performance to be negatively affected when conditioning the encoder on the target language. When the source is conditioned on only the source language, the encoder generalizes better to pairs of source and target languages not encountered during training[1].

**Conditional computation for multilingual machine translation.** A massively multilingual translation (MMT) model uses the same shared model capacity to train on several translation directions simultaneously. While doing so can lead to beneficial cross-lingual transfer between related languages, it can also add to the risk of interference between unrelated languages[1,61]. MoE models are a type of conditional computational models[62,63] that activate a subset of model parameters per input, as opposed to dense models that activate all model parameters per input. MoE models unlock marked representational capacity while maintaining the same inference and training efficiencies in terms of FLOPs compared with the core dense architecture.

However, as we increase the model capacity and the computational cost per update, the propensity for low or very low-resource languages to overfit increases, thus causing performance to deteriorate. In this section, we examine how we can use Sparsely Gated Mixture of Experts models[2–7] to achieve a more optimal trade-off between cross-lingual transfer and interference and improve performance for low-resource languages.

**Sparsely gated mixture of experts.** To build our MoE models, we substitute a quarter of the encoder and decoder feed-forward network layers with MoE layers, each with $E$ distinct experts. We followed the Top-$k$-Gating algorithm in ref. 4 and dispatched each token to at most $k = 2$ experts. For more details on the training of MoE models, see Supplementary Information E.

**Expert output masking.** In this proposed regularization strategy, we masked the expert output for a random fraction ($p_{eom}$) of the input tokens. For input tokens with dropped expert outputs, the first and/or second expert is effectively skipped. As shown in the second panel of Extended data Fig. 2, we masked both experts for the first token ($x_1$ in red), chose not to mask any of the expert outputs for the second token ($x_2$ in blue) and in the final scenario, masked only one expert for the last token ($x_3$ in green).

**Curriculum learning for MMT.** Orthogonal to model-side regularization methods such as dropout, we explored regularizing MMT models by means of CL. We proposed starting training with high-resource pairs first, then introducing low-resource pairs—prone to overfitting—in later phases. To derive the phases of the curriculum, we first trained a vanilla MoE model (without CL), followed by partitioning the translation directions into $n$ bins $\{b_1, ..., b_n\}$. If $T$ is the total number of training updates, we introduced each bin $b_i$ after $T - k_i$ updates. We based when $(k_i)_i$ and what $(b_i)_i$ directions to add at every phase of the step when we observed a language pair starting to overfit. Review the step-based CL algorithm in ref. 64 for more on how the directions are partitioned. See Supplementary Information E.2 for the list of directions added at each stage.

## Evaluations

**Automatic evaluation.** Many automatic translation quality assessment metrics exist, including model-based ones such as COMET[65] and BLEURT[66]. Although model-based metrics have shown better correlation with human judgement in recent metrics shared tasks of the WMT[43], they require training and are not easily extendable to a large set of low-resource languages. In this work, we rely on BLEU (and a variant of it) and chrF++. Both measures draw on the idea that translation quality can be quantified based on how similar a machine translation output is compared with that produced by a human translator.

**BLEU and spBLEU.** The BLEU score[44] has been the standard metric for machine translation evaluation since its inception two decades ago. It measures the overlap between machine and human translations by combining the precision of 1-grams to 4-grams with a brevity penalty. The main disadvantage of BLEU is that it is tokenization-dependent. Efforts such as sacrebleu[67] have taken strides towards standardization, supporting the use of community-standard tokenizers under the hood. However, these tokenizers do not extend to many languages. Reference 41 proposes spBLEU, a BLEU metric based on a standardized SentencePiece model (SPM) covering 101 languages, released alongside FLORES-101. In this work, we provide SPM-200 along with FLORES-200 to enable the measurement of spBLEU. (Our analyses demonstrate that there are minor differences between SPM-200 from FLORES-200 and SPM-100 from FLORES-101 when measuring on the FLORES-101 languages. The major advantage of SPM-200 is that it covers 200 languages. More details on SPM-200 are reported in section 8.1.1 of ref. 34).

**chrF++.** The chrF++ score[38] overcomes the limitation of the BLEU score, which requires that a sentence can be broken up into word tokens. However, some languages, such as Chinese or Thai, do not use spaces to separate words, and word segmentation tools may not be readily available. There is also a concern about highly agglutinative languages in which BLEU fails to assign any credit to morphological variants. chrF++ overcomes these weaknesses by basing the overlap calculation on character-level $n$-grams $F$-score ($n$ ranging from 1 to 6) and complementing with word unigrams and bi-grams. In this work, we primarily evaluated using chrF++ using the settings from sacrebleu. However, when comparing with other published work, we used BLEU and spBLEU where appropriate.

**Human evaluation methodology.** When building machine translation systems for thousands of different language pairs, a core question is which pairs reach certain levels of quality. Therefore, we needed meaningful scores that are comparable across language pairs.

**XSTS evaluation protocol.** We adapted the recently proposed XSTS methodology[48]. In short, XSTS is a human evaluation protocol focusing on meaning preservation above fluency. See details on this protocol in Supplementary Information F. For low-resource languages, translations are usually of poorer quality, and so we focused more on usable (that is, meaning-preserving) translations, even if they are not fully fluent. Compared with Direct Assessment[68] with a 5-point scale (the original direct assessment uses a 100-point scale), it is found that XSTS yields higher inter-annotator agreement[47]. XSTS rates each source sentence and its machine translation on a 5-point scale, in which 1 is the lowest and 5 is the highest.

**Calibration set.** To enable meaningful scores comparable across language pairs, we asked each evaluator to provide assessments using the XSTS scale on precisely the same set of sentence pairs. This aims to identify annotators who have a systematic tendency to be more harsh or generous in their scoring and correct for this effect. The calibration set consists of the machine translation output paired with the reference translation only in English. Based on how evaluators used the XSTS scale on this calibration set, we adjusted their raw scores on the actual evaluation task to ensure consistency across evaluators. Although this monolingual calibration task does not precisely mimic the bilingual

XSTS task, it is a reasonable first approximation and has been shown to increase the correlation between human and automatic metrics primarily by reducing one source of 'noise' in the human evaluations—the lack of score calibration between annotators.

**Obtaining aggregated human quality metrics from multiple studies.** To obtain an aggregate human quality metric for each language direction in an evaluation study, we take the majority XSTS score (that is, mean–median score) for each sentence and average these majority scores over all evaluated sentences. In a given study, the aggregate human evaluation score for any translation direction $l_s \to l_t$ is

$$H_{l_s \to l_t} = \frac{1}{|\mathcal{T}_{l_s \to l_t}|} \sum_{(S,T) \in \mathcal{T}_{l_s \to l_t}} \mathrm{median}\{X_{l_s \to l_t, i}(S, T) | 1 \le i \le M_{l_s \to l_t}\}, \quad (4)$$

where $l_s$ and $l_t$ denote the source language and the target language, respectively; $X_{l_s \to l_t, i}(S, T)$ denotes the XSTS score of the $i$th evaluator who evaluates sentences in a given translation direction $l_s \to l_t$ for a source sentence $S$ and a target sentence $T$; $M_{l_s \to l_t}$ denotes the total number of evaluators who evaluate the (source, translation) sentence pair $(S, T)$ for translation direction $l_s \to l_t$; $\mathcal{T}_{l_s \to l_t} = \{(S_{l_s \to l_t, k}, T_{l_s \to l_t, k}) | 1 \le k \le N_{l_s \to l_t}\}$ is the set of $N_{l_s \to l_t}$ (source, translation) sentence pairs being evaluated for translation direction $l_s \to l_t$.

Every evaluator in a given study $s$ is also asked to provide ratings for all or parts of a calibration set—$\mathcal{C}_s = \{(S_{s,k}, T_{s,k}) | 1 \le k \le K_s\}$. $S_{s,k}$ denotes the $k$th source sentence in the calibration set for an evaluation study; $s, T_{s,k}$ denotes the translated sentence corresponding to $S_{s,k}$; and $K_s = |\mathcal{C}_s|$ is the number of sentence pairs in the calibration set for an evaluation study.

For each language direction evaluated in a study, we obtained the majority score on the calibration set as follows:

$$C_{l_s \to l_t}^{(s)} = \frac{1}{|\mathcal{C}_s|} \sum_{(S,T) \in \mathcal{C}_s} \mathrm{median}\{X_{l,i}^{(s)}(S, T) | 1 \le i \le M_{l_s \to l_t}^{(s)}\}, \quad (5)$$

where $X_{l,i}^{(s)}(S, T)$ denotes the XSTS score provided by the $i$th evaluator, for the language direction $l_s \to l_t$, in study $s$, for a given source sentence $S$ and a translated sentence $T$, in the calibration set $\mathcal{C}_s$ of the study.

To obtain aggregated calibrated XSTS scores on the language direction level, we explored several different calibration methodologies. None of the calibration methods we investigated showed a marked difference in correlation with automated scores, and all calibration methodologies we explored provided superior correlation compared with uncalibrated XSTS scores. For more details on these calibration methodologies, see section 7.2 of ref. 34.

**Added toxicity detection for 200 languages.** To enable toxicity detection at scale, we used a detector based on word lists. In this section, we provide more details about our toxicity definition and describe the detector (ETOX) and associated word lists.

**Toxic content.** Owing to the subjective nature of toxicity, definitions of toxic language can vary. We included items that are commonly referred to as vulgar or profane language. (Note that vulgar or profane language is not always necessarily toxic. Some common slang, for instance, may be considered vulgar but is not necessarily toxic). Moreover, we also included items associated with depictions of pornographic content or sexual acts, some frequently used hate speech expressions and some expressions tied to bullying. We also included items, vulgar or not, referring to body parts that are commonly associated with sexual practices.

**The ETOX detector.** We started with the assumption that general-purpose machine translation systems should remain faithful to the source content and not add any toxic elements during the translation process. We define toxic elements as word tokens or short phrases present in our lists. ETOX identifies added toxicity using the following two

criteria: number of toxic items and matched or non-matched toxicity. A toxic item is considered detected if it is present in a line and surrounded by spaces or the start or end of a line. ETOX tracks the number of unique toxic items found in a line but does not count a phrase again if it has multiple occurrences. Matched toxicity indicates that the number of toxic items is the same in both the source and the translated content (that is, no added toxicity). Added toxicity is an instance of non-matched toxicity in which more toxic items are found in the translation output than in the source. For non-segmenting languages or some languages that use complex diacritics, space tokenization is insufficient to distinguish words from one another. In those cases, we used SentencePiece tokenization of both the sentence and toxicity word list.

**Toxicity-200 lists.** Lists are based on professional translations from English, which were then heuristically adapted by linguists to better serve the target language. As toxicity is culturally sensitive, attempting to find equivalents in a largely multilingual setting constitutes a challenge when starting from one source language. To address this issue, translators were allowed to forgo translating some of the source items and add more culturally relevant items.

In the initial release of the Toxicity-200 lists, the average number of items in a toxicity detection list was 271 entries, whereas the median number of entries was 143. The latter may be a better measure of central tendency than the mean average, given that languages with a rich inflectional morphology constitute extreme outliers (for example, the Czech list had 2,534 entries and the Polish list 2,004). The shortest list had 36 entries, and the longest 6,078.

## Data availability

All data generated and described in the Article and its Supplementary Information are available at GitHub (https://github.com/facebookresearch/fairseq/tree/nllb)[69] as follows. The FLORES-200 dataset contains human-translated evaluation data in 204 languages. The NLLB-Seed database contains human-translation seed training data in 39 languages (Supplementary Information I). The NLLB-MD database contains human-translated seed data in different domains in six languages to assess generalization (Supplementary Information J). The Toxicity-200 database contains wordlists to detect toxicity in 200 languages. Mined bitext database contains publicly available web data for 148 English-centric and 1,465 non-English-centric language pairs. Publicly available data used to train NLLB models with references to download the data are listed in Supplementary Table 2.

## Code availability

To make our work available to the community, we provide the following models and supporting code as resources freely available for non-commercial use, available at GitHub (https://github.com/facebookresearch/fairseq/tree/nllb)[69] as follows. The translation models cover 200 languages; the NLLB models come in multiple sizes (54.5B MoE, 3.3B and 1.3B Dense, and 1.3B and 600M distilled). The language identification models contain more than 200 languages. LASER3 comprises sentence encoders for identifying aligned bitext for 148 languages. Stopes consists of a data-mining library that can be used to process and clean monolingual data, followed by the creation of aligned bitext. Scripts to recreate our training data and training and generation scripts to reproduce our models are also included.

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

**Acknowledgements** We thank the following interns for their contributions to the project: C. Baziotis, D. Dua, A. Guo, O. Ignat, A. Kamran, T. Mohiuddin, A. N. Rubungo, S. Sun, S. Tan, H. Xu, S. Wu and Y. Zhang. We are grateful to all the Wikimedia Foundation staff and volunteers who worked with us and provided helpful feedback on our project. We thank V. Chaudhary for help with the data pipeline; E. Grave for his help in scaling fasttext to all FLORES-200 languages; M. Diab for her work on XSTS; L. Specia for her feedback on toxicity and XSTS; J. Ferrando and C. Escolano for their help in using the ALTI+ method; G. Chang, C.-J. Wu and R. Raghavendra for helping us to compute the CO₂ cost of training our models; A. Sridhar for helping with FSDP; S. Jeschonek, G. Anantharaman, D. Sarina, J. Colombo, S. Krishnan, D. Kannappan, K. Saladi, V. Pai, A. Yajurvedi and S. Sengupta for their assistance with training infrastructure; K. Johnson for his help with UXR studies and model evaluation; B. O'Horo and J. Kao for their generative insights and guidance; P. Fung, N. Usunier, S. Riedel, S. Sengupta and E. Dinan for their helpful feedback on the paper. We would also like to thank A. Bordes, M. Zannoli and C. Moghbel for their overall support of this project. Finally, we are indebted to the translators, reviewers, human evaluators, linguists, as well as those in the research and quality assurance agencies we partnered with, for helping to create FLORES-200, NLLB-Seed, NLLB-MD and Toxicity-200; performing human evaluations; and teaching us about their native languages.

**Author contributions** B.A., P.A., O.Ç., K. Heafield, K. Heffernan, S.J., H.S. and G.W. contributed to the data workstream of the project, which includes developing tools to facilitate data mining, cleaning and consolidation. L.B., S.B., J.C., M.E., V.G., J.M., K.R.S., A.S. and C.T. conducted research and experiments that gave rise to the models in this work. M.R.C., C.G., J.H., E.K., P.K., D.L., D.R., S.Spruit., S.W. and A.Y. implemented automatic and human evaluations of NLLB, including but not limited to quality, bias and toxicity. G.M.G., P.H., J.L. and C.R. performed all linguistics work in this project. N.F.A., S.E., A.F., F.G., A.M., S.S. and J.W. provided crucial technical and organizational leadership to help materialize this overall project. M.R.C., C.R., M.E. and S.W. prepared the paper for publication.

**Competing interests** The authors declare no competing interests.

**Additional information**
**Correspondence and requests for materials** should be addressed to Marta R. Costa-jussà.

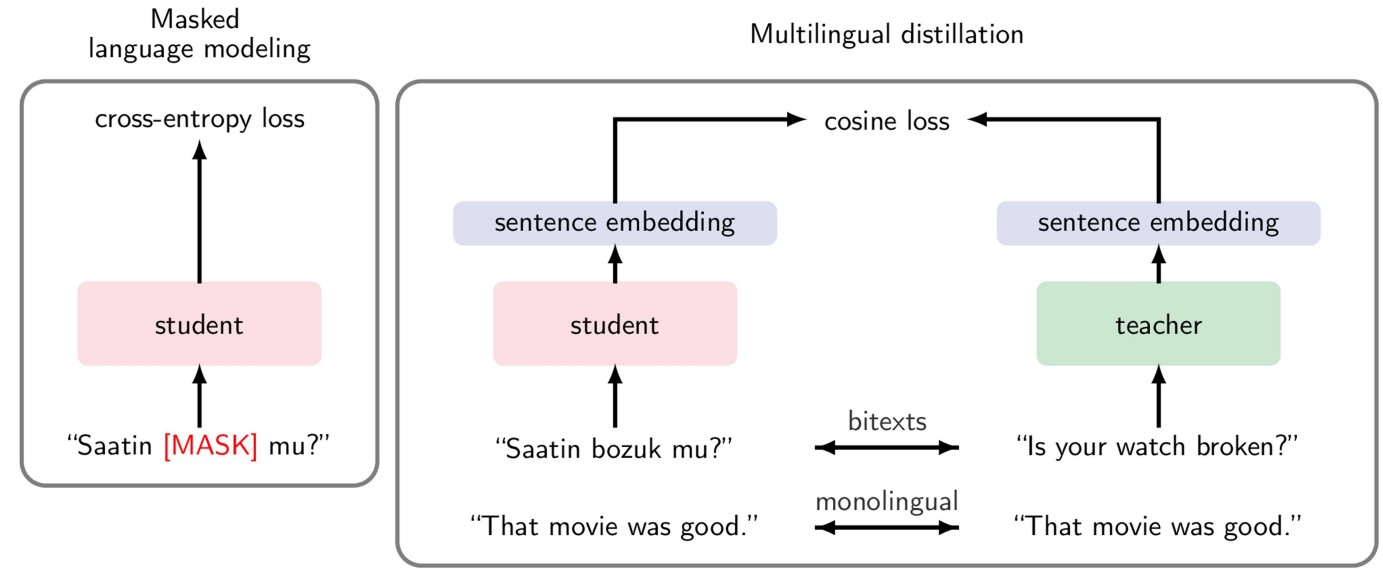

**Extended Data Fig. 1 | Architecture of the LASER3 teacher-student approach.** See[21] for more details.

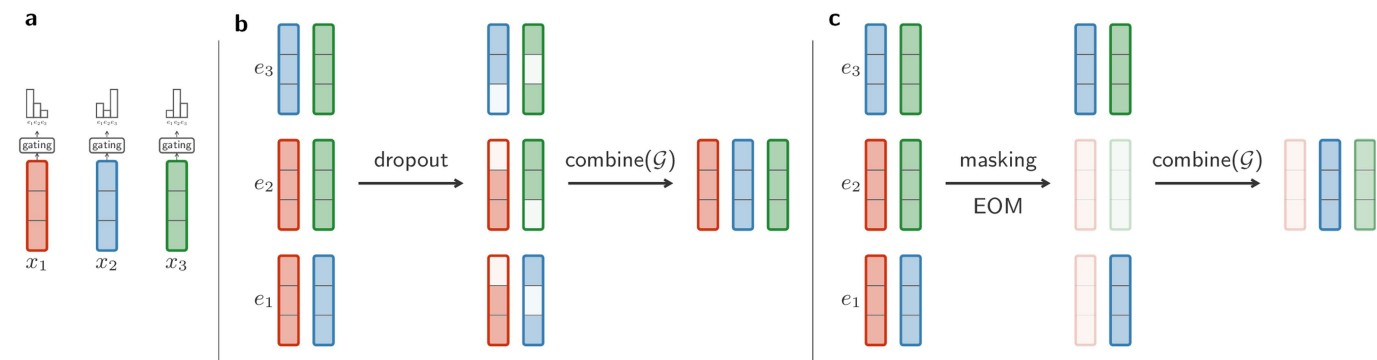

**Extended Data Fig. 2 | Illustration of EOM (panel c) in contrast to overall dropout (panel b) for MoE layers.** A color represents a token, and each token is dispatched to two experts (Top-2-Gating) depending on the gating decision (panel a). Faded colors correspond to dropped units or masked outputs.