## [Peer Review File · Nature]

Manuscript Title: Scaling Human-Centered Machine Translation

Reviewer Comments & Author Rebuttals

Reviewer Reports on the Initial Version:

Referees' comments:

Referee #1 (Remarks to the Author):

No Language Left Behind: Scaling Human-Centered Machine Translation

A. Summary of the key results:

1) The creation of Flores-200 -- a human-translated multi-parallel benchmark evaluation dataset for machine translation (MT) in 204 languages to encourage the development of massively multilingual MT model in both high-resource and low-resource languages. This is the largest evaluation dataset that has ever been created for MT. The benchmark consists of three splits: a development set (dev), an open test set (devtest), and a hidden test set (test).

2) The development of NLLB model -- a single massively multilingual MT model that covered over 200 languages. Scaling of the number of languages comes with its own challenges such as overfitting on high-resource languages and performance degradation with more languages. They proposed improved architectural design of the Sparsely Gated Mixture of Experts models and other training optimisations. They also provided several sizes of the NLLB model: 54B, 3.3B, 1.3B, and 600M parameters making it feasible for enterprises, small businesses and academic labs to run the models with even consumer GPUs.

3) The development of several tools to reproduce the models like language identification (LID) for 200 languages, LASER3 for automatically aligning bitexts from large monolingual texts of 148 languages, scripts for recreating training data and code to train the NLLB model.

4) The developing of Toxicity-200 a wordlists to detect toxicity in 200 languages, this is very important to reduce the generation of vulgar and profane texts by the NLLB models, and has a lot of applications in other domains.

5) The development of other human generated datasets such as NLLB-Seed (a small sized training data for 39 extremely low-resourced languages), NLLB-MD (an evaluation dataset in multiple domains like health, news, formal speech and informal speech), XSTS (cross-lingual semantic text similarity dataset for the evaluation of the MT outputs based on human annotation).

6) All models, scripts, and datasets are open-sourced.

B. Originality and significance: if not novel, please include reference

I found three aspects of this paper to be novel. The first is Flores-200 evaluation dataset on 204 languages which is currently the largest evaluation dataset for MT. This will help to improve the quality of MT models on several low-resource languages in the coming years.

The second novelty is the development of large scale training dataset for over 200 languages for building a machine translation model which includes language identification followed by aligning bitexts from monolingual data. While language identification seems like a simple task, they significantly improved the performance of the LID model trained on Wikipedia domain to generalise to web data (which comprises of several domains). Similarly, they developed LASER3 to automatically align bitext needed to train the models.

The third novelty is the development of the NLLB model to translate between 200 languages which is quite challenging. They developed a transformer model based on Sparsely Gated Mixture of Experts to minimize interference between unrelated language directions (as shown in Figure 1b), along with other data augmentation techniques such as back translation. This technique leads to a significant improvement of the NLLB model over existing MT models like M2M-100 and DeltaLM. Some evaluations shows that they are even better than some commercial systems e.g. Google Translate on some languages <https://arxiv.org/abs/2305.06897> . I think the authors can highlight this more if they have a comprehensive report on it.

C. Data & methodology: validity of approach, quality of data, quality of presentation

The methodology for dataset collection was adequately documented. They put more effort in quality control than the previous Flores-101 dataset. According to their report, "the number of languages requiring re-translation was only 10, down from 45 in FLORES-101". It is also challenging to obtain high-quality translation for some languages like Sicilian which took them 287 days to obtain human translation for 3,000 sentences.

One thing to note as rightly reported in Figure 2, the minimum acceptable quality is 90% which means there are few sentences in the benchmark dataset with quality issues. The authors did not provide how they intend to fix these issues going forward. In some cases, Flores-200 dataset for some languages have worse quality than those created by the low-resource language communities. For example, Adebara (2020) reported some issues with the translation to an African language, Yoruba <https://aclanthology.org/2022.acl-long.265/> (Table G.1) which includes issues with diacritics -- which I think it's easy to check automatically, spelling errors etc. The issue was reported in Flores-101 and it still persists in Flores-200. If there are some quality issues with such a language with over 40 million native speakers and several professional translators. It is possible that there are significant issues with some sentences translated in other languages that are even more under-resourced. While I understand that verifying for 200 languages is difficult, I think it would be important to come up with a plan to gradually

improve the quality of the benchmark, e.g. involve with different language communities to contribute to the further standardization of this benchmark evaluation sets.

D. Appropriate use of statistics and treatment of uncertainties

Yes, they provide useful statistics, however, the actual training data sizes used for each language is missing. They only reported categorisation by either low-resource (less than 1 million sentences) or high resource (≥ 1 million sentences) in Table A1. I think it is important to add this information to the table.

I also think several hyper-parameters and experimental details are missing in the draft, maybe they are in the GitHub repository. I do not think the details provided are not sufficient for reproducing the models.

E. Conclusions: robustness, validity, reliability

In conclusion, I think this is one of the best work on machine translation I have seen in few years, and has a significant impact on both the language speakers and the MT community. Some of the impact on Wikipedia article creation are discussed in the paper. Also, we now have a completely open-source MT model that is on par with commercial MT models like Google Translate and Microsoft Bing translator. They also provided scripts to further fine-tune or customise it for various users depending on their use cases.

F. Suggested improvements: experiments, data for possible revision

I feel some interesting experimental results were lost when the arXiv version (<https://arxiv.org/abs/2207.04672>) was compressed. For example, I could not find results and analysis of the generalisation to multiple domains using the NLLB-MD evaluation dataset.

Also, I think one important figure was missing... There is no Figure 24 in the draft. I could only find it in the arXiv version. Please add the figure.

G. References: appropriate credit to previous work?

To my surprise, many past related works are missing which were correctly documented in the arXiv version. For example, there are several benchmark dataset created by the WMT community and various language communities like Masakhane, and South Asian that were not discussed. While this is the largest MT evaluation dataset that has been developed, it would not be appropriate to completely ignore those past works, some of these datasets may even have better quality than Flores-200, especially the ones created by the low-resource language communities.

H. Clarity and context: lucidity of abstract/summary, appropriateness of abstract, introduction and conclusions

The draft is clear and properly written except for a few comments in D & F and G.

David Adelani

Referee #2 (Remarks to the Author):

Summary:

This paper presents the No Language Left Behind (NLLB) initiative, spanning several areas: multilingual data mining, large-scale many-to-many translation, language ID, evaluation sets, multilingual toxicity lists and detection, and evaluation metrics for the previous.

The paper makes a number of novel contributions in the form of data, metrics, and model performance, with large improvements compared to past work on average across up to 200 languages. The publicly available NLLB translation models and FLORES-200 evaluation sets have already made significant impact. Expanded language coverage for publicly available high-performing translation, sentence similarity, and LID models as well as evaluation sets is a significant step forward for the field.

However, some details which should be specified in a refereed publication are underspecified or not present; for appropriate comparison to this work in the future and to fully assess these models, these should be included (in "Suggested additions" below).

Organization:

Some sections with further details appear after relevant topics are initially discussed in the main text without forward references, making it appear there is not additional information. It would be helpful to reorganize or add forward references to link such places so it is clear where and for which topics such details are provided.

Some examples: Section 3.3.3 on page 20 which describes what constitutes toxicity and the toxicity lists created should be referenced in 2.3.2/2.3.3 on page 9-11 where the toxicity detection results are given to find necessary context, as the later section addresses natural questions in the earlier; EOM is mentioned and results/improvements given on page 6 and what EOM is is not described until 3.2.2 on page 16; the Appendices are not referenced in the main text; mention of SentencePiece tokenization occurs 2 pages before vocabulary size given.

Presentation:

- Typos: particularly on page 6

- Missing figure: Figure 24 referenced on page 9 does not exist in the manuscript
- Placeholder citation: CommonCrawl on page 3 says “citation” in place of a citation
- cld3 footnote is duplicated on page 4: use footnotemark the second time
- Language labels in Fig. 1 are unreadable and rows difficult to distinguish, making it hard to reference the in-text analysis

Suggested additions:

- LID training data is not documented or provided: “We use publicly available data sets to train our LID system” without any citations, references, or further details.
- LID “our approach combines data-driven fasttext with a set of handwritten rules...” no details on these rules. Will these be released?
- Data mining: “Languages are trained either as individual students or together with languages of the same family.” > for which languages are each approach taken? What is the performance difference? How was this decided?
- Data mining: Aggregate “>1.1 billion new sentence pairs for 148 languages” is stated. How much data is mined per-language and what is the distribution across languages? How does this relate to performance improvements per language pair?
- NLLB-Seed data: the seed data for 39 languages are mentioned by name once in the main text and not described further (composition, size, etc). NLLB-MD is mentioned only in Section 6 mentioning what is released.
- Table 2: “201 pairs” → should this be 199 pairs? (200 total minus English?)
- Table 2: languages in the appendix are broken into “high” and “low.” Which are in the “v.low” category listed in Table 2 and what are the thresholds between categories?
- 2.2: what is K?
- 2.3.1: could the instructions to annotators be provided or additional information to contextualize what a 90/100 quality threshold is? For Figure 3, what is the scale above 90? (i.e. it is not clear what the highest score on this scale is)
- FastText model: buckets of 1M – how many? What is the total vocabulary? What does “words” here refer to for e.g. non-whitespace marking languages?
- SentencePiece: in addition to the toolkit, cite the method paper (Kudo 2018: <https://aclanthology.org/P18-1007/>). State whether using the unigramLM vs BPE algorithm.
- “Such a large vocabulary size ensures adequate representation across the wide spectrum of languages we support.” → This is not necessarily the case, particularly for character-based languages such as Chinese (ex <https://github.com/facebookresearch/fairseq/issues/4560>), and this should likely be discussed further or rephrased.
- Curriculum 3.2.3: ‘base when and what directions to add at every phase based on the step where we observe overfitting.’ > what is your criteria for overfitting? Details on how the translation directions partitions are decided (n, b) and when they are to be added (k) would be nice to see here. A forward reference to the appendix listing the final model’s lang buckets would be helpful.
- Toxicity lists: what is the distribution of the number of entries? Can the sizes per language be added to the appendix? For Table 4, in addition to the number of toxic items added in each setting, it would be

nice to see the size of the lists used for that pair for context

- There is significant literature on vocabulary challenges for multilingual translation models, and particularly for low-resource languages, but the only citations here are from two 2022 papers. This should likely be expanded to include related additional work (select examples:

<https://openreview.net/pdf?id=Skeke3C5Fm>, <https://aclanthology.org/N18-1032/>,
<https://aclanthology.org/2020.emnlp-main.367/>, <https://arxiv.org/abs/2301.10472>).

- The number of languages in the world is typically stated to be >7,000 rather than assigned a specific value because divisions between languages and dialects are not agreed upon. Suggest changing 7151 → >7,000

- What are the differences in performance between the different model scale variants provided?

- Results by language rather than solely aggregate scores would be nice to include in the appendix for comparisons to this work.

Referee #3 (Remarks to the Author):

The paper works towards a multilingual model with high representation of low resource languages. The eventual model has a combination of high and low resource languages and enables pair wide translations.

The development dataset built is a novel contribution and introduces scalability across languages to their model.

The paper also substantially expands evaluation paradigms and datasets that look at overall quality and safety in all languages represented.

The paper does extensive evaluations and analysis to support claims of representation and translation for low resource languages, and is clear in presentation.

Suggestions and questions:

1. The XSTS protocol is central to the evaluation of quality of embeddings in this work, and the human eval at scale hinges on this protocol. However, the exact setup could be more detailed on exact sentences used, how they were chosen for consistency across languages, and meaning preservation across the 200 languages considered in this work.

2. It is unclear to what extent a human annotator has agency to include local, colloquial slurs, text for toxicity evaluations that is unrelated to text shown to them at any instance. Authors indicate that open contribution from annotators was structured and details would help calibrate the degree of inclusion of languages.

3. Payment details of annotators missing and essential for publication of work engaging with humans. Distribution of authors and their demographics and location are also essential.

4. Highly recommend using a data card for each dataset created and a model card for each model created. While the authors open source their developments, their utility is limited without transparency about intended usage, etc.

Referee #4 (Remarks to the Author):

Summary of the key results

The goal of the paper is to train a Multilingual Machine Translation model across 200 languages, and so dealing with thousands of language pair directions. To achieve this ambitious objective, several challenges need to be addressed and thus consisting in key results of the paper:

1) mining multilingual data, and identifying the related languages. The authors present first a system for language identification based on the fasttext model trained over their proposed cross-lingual data, and then a bitext miner based on the teacher-student framework extending the LASER model to the 200 languages of interest.

2) Filtering data. The authors proposed to filter toxic data for such a high number of languages, ie 200, by first manually creating a list of toxic n-gram per language, and then filtering out unbalanced toxic translations, ie cases in which one sentence in the bitext pair contains at least 2 more toxic items than the other sentence in the pair.

3) modeling and training a neural architecture capable of handling such a diverse and unbalanced dataset. To address this challenge, the authors employ a Transformer architecture coupled with mixture of expert layers, designing an expert output masking strategy as a training regularizer, and a curriculum learning strategy to counterbalance the overfitting on low-resource language pairs. The general idea is that for expert output masking, an expert output is masked for a random fraction of the input tokens, while for curriculum learning, the model is first trained over high-resource pairs, then low-resource ones, and finally, language pairs are introduced in phases where, given a total number T of training steps, pairs that empirically overfit within K updates are introduced K updated before the end of the training T .

4) creating a test set to evaluate thousands of language pair directions. This work extends a previous effort that covered 101 languages, to 200, through a workflow consisting of 4 phases: alignment, translation, initial quality assurance and iterations, final quality assurance and completion. These steps were re-iterated until a human quality test achieve a 90% quality score to ensure an acceptable standard.

5) reliable human evaluation over more than 50 language pairs. Besides the use of common automatic evaluation metrics, the authors use a crosslingual semantic text similarity metric, XSTS, emphasizing meaning preservation over fluency. XSTS uses a 5-point scale where 1 means no meaning preserved and 3 is an acceptable threshold. Moreover, to ensure consistency across evaluators, the authors use a calibration set, asking each annotator to provide an assessment on this set using the XSTS scale. Finally, to aggregate the human quality metric, the majority of the XSTS score for each sentence is taken, and then averaged over all sentences.

6) Last but not least, the paper open sources all the contributions, including the model, the data, the benchmarks, and the code.

Originality and significance: if not novel, please include reference

The methods used throughout the paper are not novel per se, as also the authors give appropriate credits through the references for each section, the main contribution is the new testing data covering other 100 languages not available before, scaling the training of the model to such a big number of languages, obtaining better results than prior state of the art models, and releasing everything to the benefit of the community.

Data & methodology: validity of approach, quality of data, quality of presentation

Both the proposed new test data and the protocol on how the MT system has been trained are valid, even though I am not sure about data contamination, ie whether the training data do not contain test data as well. I guess the authors checked for it, but i did not find any explicit reference on this submission.

Appropriate use of statistics and treatment of uncertainties

The metrics are valid and appropriate given the addressed task. Even though there are missing details about the annotators for the FLORES-200 test data proposed. How they have been instructed, how many were they, which platform they used for doing their task, if they are/were employees or hired through some other external platform, which was their level of education, and so on.

Conclusions: robustness, validity, reliability

The paper concludes by mentioning the impact of the availability of their models and data on the community, both academic and industrial, reducing the digital language barrier on the web. I do agree with the impact that this work will/already have.

Suggested improvements: experiments, data for possible revision

References: appropriate credit to previous work?

Clarity and context: lucidity of abstract/summary, appropriateness of abstract, introduction and conclusions

I am condensing these 3 last points in one, as i know that exist a almost 200 pages manuscript version of this manuscript, and squishing the size to 30 pages is a hard task, and most probably my potential suggestions are already there in the longer version.

The clarity of this shortened submission could have been improved, as for example having an introduction, results, and then method as sections is not so common and intuitive for a reader.

Finally, why did not you submit to TACL instead?

Author Rebuttals to Initial Comments:

Referee #1

COMMENT: One thing to note as rightly reported in Figure 2, the minimum acceptable quality is 90% which means there are few sentences in the benchmark dataset with quality issues. The authors did not provide how they intend to fix these issues going forward. In some cases, Flores-200 dataset for some languages have worse quality than those created by the low-resource language communities. For example, Adebara (2020) reported some issues with the translation to an African language, Yoruba <https://aclanthology.org/2022.acl-long.265/> (Table G.1) which includes issues with diacritics – which I think it's easy to check automatically, spelling errors etc. The issue was reported in Flores-101 and it still persists in Flores-200. If there are some quality issues with such a language with over 40 million native speakers and several professional translators. It is possible that there are significant issues with some sentences translated in other languages that are even more under-resourced. While I understand that verifying for 200 languages is difficult, I think it would be important to come up with a plan to gradually improve the quality of the benchmark, e.g. involve with different language communities to contribute to the further standardization of this benchmark evaluation sets.

RESPONSE: While we checked the quality of the datasets (beyond the standard practice in standard competitions like WMT), the more rigorous Q/A process is only done over a statistical sample (20%) of the data (due to high costs). In addition, we only sent the dataset to full rework when the amount of errors exceeds 10%. Otherwise, detected errors are fixed on the spot. This, of course, leaves the door open for some mistakes in the data. However, we do our best to act on feedback from the community about systematic errors. For example, issues with languages like Sinhala in FLORES-101, where inconsistencies in tokenization have been documented by the community, have been taken into account in FLORES-200. We will continue to monitor publications or feedback from researchers and other language communities to improve future iterations of our work.

D. Appropriate use of statistics and treatment of uncertainties

COMMENT: Yes, they provide useful statistics, however, the actual training data sizes used for each language is missing. They only reported categorisation by either low-resource (less than 1 million sentences) or high resource (≥ 1 million sentences) in Table A1. I think it is important to add this information to the table.

RESPONSE: These details are in the preprint and we added a reference to where the reader can find these details. For further details on the task set-up, such as the amount of training data per language pair, the reader can refer to appendix E or the Section 8 (NLLB team et al. 2022).

COMMENT: I also think several hyper-parameters and experimental details are missing in the draft, maybe they are in the GitHub repository. I do not think the details provided are not sufficient for reproducing the models.

RESPONSE: We have now added more details pertaining to the reviewer's request, but specific details on how to reproduce the models we describe requires descriptions we do not have words for given space constraints. Interested readers can refer to our preprint for more of such details.

F. Suggested improvements: experiments, data for possible revision

COMMENT: I feel some interesting experimental results were lost when the arXiv version (<https://arxiv.org/abs/2207.04672>) was compressed. For example, I could not find results and analysis of the generalisation to multiple domains using the NLLB-MD evaluation dataset.

RESPONSE:

Maha: We can add to the appendix experimental results with finetuning on NLLB-MD Added in Appendix D.3.

COMMENT: Also, I think one important figure was missing... There is no Figure 24 in the draft. I could only find it in the arXiv version. Please add the figure.

RESPONSE: We have now added the figure.

G. References: appropriate credit to previous work?

COMMENT: To my surprise, many past related works are missing which were correctly documented in the arXiv version. For example, there are several benchmark dataset created by the WMT community and various language communities like Masakhane, and South Asian that were not discussed. While this is the largest MT evaluation dataset that has been developed, it would not be appropriate to completely ignore those past works, some of these datasets may even have better quality than Flores-200, especially the ones created by the low-resource language communities.

RESPONSE: We thank the reviewer for noting this and have added more citations (Kuwanto et al., 2021; Nekoto et al., 2020; Orife et al., 2020), in addition to some others that were listed in the preprint version. We would add more if we could but are constrained by the number of references.

Referee #2

COMMENT: Some sections with further details appear after relevant topics are initially discussed in the main text without forward references, making it appear there is not

additional information. It would be helpful to reorganize or add forward references to link such places so it is clear where and for which topics such details are provided. Some examples: Section 3.3.3 on page 20 which describes what constitutes toxicity and the toxicity lists created should be referenced in 2.3.2/2.3.3 on page 9-11 where the toxicity detection results are given to find necessary context, as the later section addresses natural questions in the earlier; EOM is mentioned and results/improvements given on page 6 and what EOM is is not described until 3.2.2 on page 16; the Appendices are not referenced in the main text; mention of SentencePiece tokenization occurs 2 pages before vocabulary size given.

RESPONSE: We have rectified these issues and added the necessary signposts.

COMMENT:

- Typos: particularly on page 6
- Missing figure: Figure 24 referenced on page 9 does not exist in the manuscript
- Placeholder citation: CommonCrawl on page 3 says "citation" in place of a citation
- cld3 footnote is duplicated on page 4: use footnotemark the second time
- Language labels in Fig. 1 are unreadable and rows difficult to distinguish, making it hard to reference the in-text analysis

RESPONSE: We have addressed the first four errors. For Figure 1, the reader can zoom in and it allows them to see languages in their completeness. We want to give the reader a general visual representation of the results (and zooming in could help)

COMMENT: - LID training data is not documented or provided: "We use publicly available data sets to train our LID system" without any citations, references, or further details

RESPONSE: We have added the necessary references to datasets (Commoncrawl) and also the specific sections in the NLLB preprint.

COMMENT: LID "our approach combines data-driven fasttext with a set of handwritten rules..." no details on these rules. Will these be released?

RESPONSE: We added references and more details on the linguistic rules.

COMMENT: Data mining: "Languages are trained either as individual students or together with languages of the same family." > for which languages are each approach taken? What is the performance difference? How was this decided?

RESPONSE: We added a reference to the training of the students, which follows the approach described in greater detail in Heffernan et al. (2022)

COMMENT: Data mining: Aggregate “>1.1 billion new sentence pairs for 148 languages” is stated. How much data is mined per-language and what is the distribution across languages? How does this relate to performance improvements per language pair?

RESPONSE: We have now added more information on this in appendix C.

COMMENT: NLLB-Seed data: the seed data for 39 languages are mentioned by name once in the main text and not described further (composition, size, etc). NLLB-MD is mentioned only in Section 6 mentioning what is released.

RESPONSE: We have added an appendix on this (G).

COMMENT: Table 2: “201 pairs” – should this be 199 pairs? (200 total minus English?)

RESPONSE: Done.

COMMENT: Table 2: languages in the appendix are broken into “high” and “low.” Which are in the “v.low” category listed in Table 2 and what are the thresholds between categories?

RESPONSE: We added a note to Table 2 to say that a language is defined as very low-resource if it has fewer than 100k samples across all pairings with any other language in our dataset.

COMMENT: 2.2: what is K?

RESPONSE: k -th source sentence in the calibration set for the evaluation study.

COMMENT: - 2.3.1: could the instructions to annotators be provided or additional information to contextualize what a 90/100 quality threshold is? For Figure 3, what is the scale above 90? (i.e. it is not clear what the highest score on this scale is)

RESPONSE: We added a new appendix (C) to provide more details on what the evaluators were tasked to do: “The final human quality test encompassed a 20% assessment by independent reviewers from a language service provider (LSP). The reviewers assessed for translation errors on the sentence level, and the translation quality score per language was determined based on the number of errors identified by the reviewers. The following errors were examined: grammar, punctuation, spelling, capitalization, addition or omission of information, mistranslation, unnatural translation, untranslated text, and register. Each error is also associated with a severity level—minor, major, and critical. The overall score is constructed by tallying these different error types. The acceptable translation quality score

was set at 90%. It is also important to note that there was first an initial alignment between the translators and LSP on the approach to take for each language. In cases of large disagreements, translators were also allowed to arbitrate with the reviewers to further align their understanding of translation quality. This was especially helpful for languages with lower levels of standardization.”

For Figure 3, the highest score is 100%, 92% means independent raters agreed with 92% of the FLORES reference translations in that direction. We added a clarification to the paper on this.

COMMENT: FastText model: buckets of 1M – how many? What is the total vocabulary? What does “words” here refer to for e.g. non-whitespace marking languages?

RESPONSE: In fastText, we say “word” when it is separated by spaces. So when it’s a non-whitespace marking language, there is only one “word” for the whole sentence, but we take character n-grams; the total vocabulary is the buckets + number of “word”s the character n-grams are bucketed inside the 1M buckets, and the “words” are kept in the vocabulary the words are kept if they appear more than 1000 times. All details on FastText are reported in section 5.1.2 (NLLB Team et al. 2022)

COMMENT:- SentencePiece: in addition to the toolkit, cite the method paper (Kudo 2018: <https://aclanthology.org/P18-1007/>). State whether using the unigramLM vs BPE algorithm.

RESPONSE: Citation has been added. Further details of SPM are in section 8.1.1 (NLLB Team et al. 2022). You can see more details in footnote 12.

COMMENT: - “Such a large vocabulary size ensures adequate representation across the wide spectrum of languages we support.” - This is not necessarily the case, particularly for character-based languages such as Chinese (ex <https://github.com/facebookresearch/fairseq/issues/4560>), and this should likely be discussed further or rephrased.

RESPONSE: Subword-level tokenization is the sota tokenization in most NLP tasks; this is the reason we don’t use character-level tokenization. Besides, even if character-level tokenization works best for some set of languages, our goal is to train a single multilingual MT model with a shared vocabulary. The issue with Chinese characters in the NLLB tokenizer is an artifact of data sampling when learning the vocabulary with a BPE algorithm. The argument about the need for a large vocabulary when covering a large set of languages is still valid. When learning smaller vocabularies, a large amount of tokens/sub-words will be substituted with the unknown token (<UNK>). The vocabulary size was chosen to

minimize the percentage of these generated <UNK> tokens when tokenizing our training data.

COMMENT: - Curriculum 3.2.3: 'base when and what directions to add at every phase based on the step where we observe overfitting.' > what is your criteria for overfitting? Details on how the translation directions partitions are decided (n, b) and when they are to be added (k) would be nice to see here. A forward reference to the appendix listing the final model's lang buckets would be helpful.

RESPONSE: We wrote a detailed description of bucketing in <https://aclanthology.org/2023.findings-acl.897/> (Algorithm 1). There are buckets in the appendix C, which we refer to this.

COMMENT: - Toxicity lists: what is the distribution of the number of entries? Can the sizes per language be added to the appendix? For Table 4, in addition to the number of toxic items added in each setting, it would be nice to see the size of the lists used for that pair for context

RESPONSE: Toxicity lists are available online, so the reader can refer to all this information directly (<https://github.com/facebookresearch/flores/blob/main/toxicity/README.md>).

COMMENT: There is significant literature on vocabulary challenges for multilingual translation models, and particularly for low-resource languages, but the only citations here are from two 2022 papers. This should likely be expanded to include related additional work (select examples: <https://openreview.net/pdf?id=Skeke3C5Fm>, <https://aclanthology.org/N18-1032/>, <https://aclanthology.org/2020.emnlp-main.367/>, <https://arxiv.org/abs/2301.10472>).

RESPONSE: We thank the reviewer for these and have added all of these citations to our work.

COMMENT: The number of languages in the world is typically stated to be >7,000 rather than assigned a specific value because divisions between languages and dialects are not agreed upon. Suggest changing 7151 → >7,000

RESPONSE: We changed this to 7,000+ languages.

COMMENT: What are the differences in performance between the different model scale variants provided?

RESPONSE: We share all metrics in the github repo (<https://github.com/facebookresearch/fairseq/tree/nllb>). We evaluated almost 40K directions. Adding those to a table may not be feasible for this publication. We can add English-centric directions if needed.

COMMENT: Results by language rather than solely aggregate scores would be nice to include in the appendix for comparisons to this work.

RESPONSE: See response above.

Referee #3

Suggestions and questions:

COMMENT: The XSTS protocol is central to the evaluation of quality of embeddings in this work, and the human eval at scale hinges on this protocol. However, the exact setup could be more detailed on exact sentences used, how they were chosen for consistency across languages, and meaning preservation across the 200 languages considered in this work.

RESPONSE: Due to space constraints, we added what we could to the current revision, but refer readers to more details in section 8.3.3 of NLLB team et al. 2022.

COMMENT: 2. It is unclear to what extent a human annotator has agency to include local, colloquial slurs, text for toxicity evaluations that is unrelated to text shown to them at any instance. Authors indicate that open contribution from annotators was structured and details would help calibrate the degree of inclusion of languages. Payment details of annotators missing and essential for publication of work engaging with humans. Distribution of authors and their demographics and location are also essential.

RESPONSE: Details about annotators are now it is added in appendix E. Also, calibration details are referenced from the main paper in the same appendix.

COMMENT: Highly recommend using a data card for each dataset created and a model card for each model created. While the authors open source their developments, their utility is limited without transparency about intended usage, etc.

RESPONSE: We have added models and datasets in the appendix.

Referee #4

Data & methodology: validity of approach, quality of data, quality of presentation

COMMENT: Both the proposed new test data and the protocol on how the MT system has been trained are valid, even though I am not sure about data contamination, ie whether the training data do not contain test data as well. I guess the authors checked for it, but i did not find any explicit reference on this submission.

RESPONSE: We thank the reviewer for noting this. I added a footnote on how we filtered out FLORES validation and test data as well as any other benchmarks we evaluated on (WMT, IWSLT, etc.). This was done by comparing the hashes of training sentences against those of evaluation sentences, using the xxHash algorithm.

COMMENT:I am condensing these 3 last points in one, as i know that exist a almost 200 pages manuscript version of this manuscript, and squishing the size to 30 pages is a hard task, and most probably my potential suggestions are already there in the longer version. The clarity of this shortened submission could have been improved, as for example having an introduction, results, and then method as sections is not so common and intuitive for a reader.

RESPONSE: We followed Nature's guidelines in our reformatting effort and have made significant effort in increasing the readability and flow of the article in this version.

Reviewer Reports on the First Revision:

Referees' comments:

Referee #1 (Remarks to the Author):

A. Summary of the key results:

1) The creation of Flores-200 -- a human-translated multi-parallel benchmark evaluation dataset for machine translation (MT) in 204 languages to encourage the development of massively multilingual MT model in both high-resource and low-resource languages. This is the largest evaluation dataset that has ever been created for MT. The benchmark consists of three splits: a development set (dev), an open test set (devtest), and a hidden test set (test).

2) The development of NLLB model -- a single massively multilingual MT model that covered over 200 languages. Scaling of the number of languages comes with its own challenges such as overfitting on high-resource languages and performance degradation with more languages. They proposed improved architectural design of the Sparsely Gated Mixture of Experts models and other training optimisations. They also provided several sizes of the NLLB model: 54B, 3.3B, 1.3B, and 600M parameters making it feasible for enterprises, small businesses and academic labs to run the models with even consumer GPUs.

3) The development of several tools to reproduce the models like language identification (LID) for 200 languages, LASER3 for automatically aligning bitexts from large monolingual texts of 148 languages, scripts for recreating training data and code to train the NLLB model.

4) The developing of Toxicity-200 a wordlists to detect toxicity in 200 languages, this is very important to reduce the generation of vulgar and profane texts by the NLLB models, and has a lot of applications in other domains.

5) The development of other human generated datasets such as NLLB-Seed (a small sized training data for 39 extremely low-resourced languages), NLLB-MD (an evaluation dataset in multiple domains like health, news, formal speech and informal speech), XSTS (cross-lingual semantic text similarity dataset for the evaluation of the MT outputs based on human annotation).

6) All models, scripts, and datasets are open-sourced.

B. Originality and significance: if not novel, please include reference

Same as the last review

C. Data & methodology: validity of approach, quality of data, quality of presentation

Authors have addressed the comment on how to improve mistakes in the dataset.

D. Appropriate use of statistics and treatment of uncertainties

I had some comments on missing experimental details. Authors have provided a few more experimental details in the updated draft.

E. Conclusions: robustness, validity, reliability

In conclusion, I think this is one of the best work on machine translation I have seen in few years, and has a significant impact on both the language speakers and the MT community. Some of the impact on Wikipedia article creation are discussed in the paper. Also, we now have a completely open-source MT model that is on par with commercial MT models like Google Translate and Microsoft Bing translator. They also provided scripts to further fine-tune or customise it for various users depending on their use cases.

F. Suggested improvements: experiments, data for possible revision

I had comment on experimental results on multiple domains. Glad the authors added them in appendix.

Other issue with missing figure has been addressed.

There is one missing citation on page 45, before appendix F (This is particularly relevant when finetuning on a single task for which NLLB- 200 has learned to assign specific experts (see ??)). Please fix this.

G. References: appropriate credit to previous work?

I had some comments about missing related works, authors have added some references for African and south-east Asian languages. The issue has been addressed to a certain extent.

H. Clarity and context: lucidity of abstract/summary, appropriateness of abstract, introduction and conclusions

David Adelani

Referee #3 (Remarks to the Author):

The authors have provided additional details on the dataset, the model cards, the test/train separation, human evaluation details and split of the annotator pool, and have rearranged writing for overall readability. Combining the author response to my previous review, and the updated manuscript, I think the paper is in great shape and should be published. In the rapidly evolving climate of LLM development and deployment, this effort towards meaningful and effective multilinguality is very valuable and will help improve access issues across the globe.

Referee #4 (Remarks to the Author):

The authors have addressed most of the reviewers' suggestions, and the paper is now improved with respect to the first version. Some typos still remain, such as page12 "4 summarizes" (it misses the figure), page17 "section e.2" (it's an appendix).

I would also suggest to the authors to include in Section 6, in supplementary information, a bullet point related to model card appendixes, and to at least mention, in the conclusion, related initiatives such as [1,2,3].

[1] Tiedemann J, Aulamo M, Bakshandaeva D, Boggia M, Grönroos SA, Nieminen T, Raganato A, Scherrer Y, Vazquez R, Virpioja S. Democratizing Machine Translation with OPUS-MT. arXiv preprint arXiv:2212.01936. 2022 Dec 4.

[2] Tiedemann J. The Tatoeba Translation Challenge--Realistic Data Sets for Low Resource and Multilingual MT. In Proceedings of the Fifth Conference on Machine Translation 2020 Nov (pp. 1174-1182).

[3] Tiedemann J, Thottingal S. OPUS-MT--Building open translation services for the World. In Proceedings of the 22nd Annual Conference of the European Association for Machine Translation 2020 Nov 1. European Association for Machine Translation.

Author Rebuttals to First Revision:

Response to referees

Reviewer 1:

COMMENT: There is one missing citation on page 45, before appendix F (This is particularly relevant when finetuning on a single task for which NLLB- 200 has learned to assign specific experts (see ??)). Please fix this.

ANSWER: Thanks for pointing this out, we have now added the specific reference.

Referee #4 (Remarks to the Author):

COMMENT: Some typos still remain, such as page12 “4 summarizes” (it misses the figure), page17 “section e.2” (it’s an appendix).

ANSWER: Solved.

COMMENT: I would also suggest to the authors to include in Section 6, in supplementary information, a bullet point related to model card appendixes, and to at least mention, in the conclusion, related initiatives such as [1,2,3].

[1] Tiedemann J, Aulamo M, Bakshandaeva D, Boggia M, Grönroos SA, Nieminen T, Raganato A, Scherrer Y, Vazquez R, Virpioja S. Democratizing Machine Translation with OPUS-MT. arXiv preprint arXiv:2212.01936. 2022 Dec 4.

[2] Tiedemann J. The Tatoeba Translation Challenge—Realistic Data Sets for Low Resource and Multilingual MT. In Proceedings of the Fifth Conference on Machine Translation 2020 Nov (pp. 1174-1182).

[3] Tiedemann J, Thottingal S. OPUS-MT--Building open translation services for the World. In Proceedings of the 22nd Annual Conference of the European Association for Machine Translation 2020 Nov 1. European Association for Machine Translation.

ANSWER: Thanks! The bullet point related to model card is now added in Supplementary information. We appreciate the suggestion regarding the inclusion of additional references. However, due to the formatting requirements of the Nature journal, we have a limited number of references we can include in this section. Since the suggested references are not strictly related to the content of the paper, we have decided to de-prioritize them. We assure you that this decision was made solely based on the constraints of the format and not on the value of the suggested references.